# Who reported having a high-strain job, low-strain job, active job and passive job? The WIRUS Screening study

Tore Bonsaksen [1,2]*, Mikkel Magnus Thørrisen[1,3], Jens Christoffer Skogen[3,4,5], Randi Wågø Aas[1,3,6]*

**1** Department of Occupational Therapy, Prosthetics and Orthotics, Faculty of Health Sciences, OsloMet– Oslo Metropolitan University, Oslo, Norway, **2** Faculty of Health Studies, VID Specialized University, Sandnes, Norway, **3** Faculty of Health Sciences, University of Stavanger, Stavanger, Norway, **4** Department of Health Promotion, Norwegian Institute of Public Health, Bergen, Norway, **5** Alcohol & Drug Research, Stavanger University Hospital, Stavanger, Norway, **6** Presenter–Making Sense of Science, Stavanger, Norway

* tore.bonsaksen@oslomet.no (TB); randi.aas@uis.no (RWA)

## Abstract

### Objective

The Job Demands and Control model classifies job types as active, passive, low-strain or high-strain, based on a combination of job demands and control. While studies have shown high-strain jobs to have adverse consequences for health and work participation, prognostic factors for the four job types have been less explored. The aim of this study was to assess the associations between sociodemographic factors and job descriptors and being in high-strain, low-strain, active and passive jobs.

### Methods

The WIRUS Screening study targeted Norwegian employees in private and public enterprises. In this study, associations with job types among 4,487 employees were investigated with binary logistic regression analyses, adjusting for sociodemographic and job-related variables.

### Results

In fully adjusted models, high-strain job was associated with female gender; lower education; shift work; and doing work outside the workplace. Low-strain job was associated with opposite scores on the same variables, and with lower job position. Active job was associated with lower age; female gender; higher levels of education; higher job position level; shift work; and not doing work outside the workplace. Passive job was associated with opposite scores on the same variables.

**Data Availability Statement:** The data from the study contain potentially sensitive information. In accordance with restrictions imposed by the Regional Committees for Medical and Health

Research in Norway (approval no. 2014/647), data must be stored on a secure server at the University of Stavanger. The contents of the ethics committee's approval resolution as well as the wording of participants' written consent do not render open public data access possible. Access to the study's minimal and depersonalized data set may be requested by contacting the Faculty of Health Sciences at University of Stavanger (post@uis.no).

**Funding:** The Norwegian Directorate of Health and the Research Council of Norway funded the study. The funder had no role in study design, data collection and analysis, decision to publish, or preparation of the manuscript.

**Competing interests:** The authors have declared that no competing interests exist.

## Conclusions

The study corroborates the role gender and education have for experiencing the job, and expands on existing knowledge on the role of job position and irregular working hours and spaces. By identifying factors associated with job types, the prevention of health problems and work disability may become be more targeted.

## Introduction

In April 2019, 69.9% of the population in Norway had employment and only 3.9% were registered as unemployed [1]. Of those who were employed, the majority of both women (63.1%) and men (85.4%) had full time employment [2], which in general sums up to 37.5 hours per week. While simply having a job has been associated with lower risk of mental health problems in general population samples [3, 4], the considerable proportion of time people spend on their job would also indicate that the work environment is an important arena for health promotion initiatives [5].

In general terms, a mentally healthy workplace has been characterized as one in which risk factors are recognized and acted on, while protective and resilience factors are maximized [6]. Perceptions of the work environment are likely to be multi-dimensional, such that careful operationalization of relevant concepts is required. The Job Demands and Control (JDC) model, originating from Karasek's early work [7], identified psychological demands and decision latitude as core aspects of the perceived work environment. While psychological demands refer to the work pressure and the workload experienced in the job, decision latitude refers to the range of skills asked for in the job and the level of control the employee has over decisions that are important to his work. The JDC model transcends traditional notions of high demands as a risk factor for poor employee health and dissatisfaction in itself, and suggests that job strain is a function of job demands interacting with the employee's level of decision latitude. The combination of the two concepts (psychological demands and decision latitude) allows for classifying jobs into four main types [8]. Active jobs, which are described as contributing positively to employees' development, health and participation in the workplace, pose high demands while also allowing the employee to have a high degree of decision latitude. Passive jobs indicate low demands in combination with low decision latitude, whereas low-strain jobs are characterized by low demands combined with a high degree of decision latitude. High-strain jobs are those characterized by high demands in combination with low levels of decision latitude among employees, which appears to constitute the most problematic job type in relation to health and related outcomes.

While relatively little research appears to concern the active, passive, and low-strain job types originating from the JDC model, the conditions that create high-strain jobs have been studied extensively (e.g., [9, 10]). Occupations frequently found to induce high strain are for example machine-paced assemblers and freight handlers, but also low-status occupations with high proportions of women, such as nurses aids and waitresses [11]. High-strain jobs have been associated with a variety of psychological well-being and stress-related outcomes [12–16], discrete medical conditions such as cardiovascular illness [17], and also sickness absence from work [18–22] and return to work after sick leave [23]. Stansfeld and Candy's [12] meta-analytic review showed that the risk of any common mental disorder was higher for persons reporting lower decision latitude and higher psychological demands. High job strain, as indicated by the combination of high psychological demands and low decision latitude, yielded the largest effect size.

More recently, the systematic review conducted by Nieuwenhuijsen and co-workers [24] concluded that there is strong evidence supporting a relationship between high psychological demands and the occurrence of stress-related disorders of clinical relevance, including diagnoses such as neurasthenia, adjustment disorders and burnout. Similarly, they found strong evidence for a relationship between low job control and stress-related disorders, and for men in particular. In Norway, the JDC scales have been examined as predictors of case-level anxiety and depression in a general population sample [25]. It was found that low control and high demands were individual risk factors for both outcomes. When combining the JDC scales, those classified with high-strain jobs had significantly higher anxiety and depression scores compared to those in the other three groups. Still, we know less about prognostic characteristics of those reporting having high-strain jobs.

In summary, a substantial amount of research points toward different health and work participation outcomes in relation to different combinations of the decision latitude and psychological demands scales, and outcomes related to the high-strain job type. However, fewer studies have been focused on identifying sociodemographic or job-related characteristics associated with the different job types. Such studies may provide indication of groups of employees who might have higher probability of being in high-strain, low-strain, active and passive jobs. In particular, the sparse amount of studies to date focusing on the active, passive and low-strain job types appears to be a missed opportunity to gain more insight into the distribution of different types of jobs across a range of sociodemographic and job-related variables. Therefore, the aim of this study was to examine sociodemographic characteristics and job descriptors and their associations with high-strain, low-strain, active and passive jobs.

## Materials and methods

### Design

This study is part of the Norwegian national WIRUS project. The present study used cross-sectional data collected between 2014 and 2018, and reported by employees from the private and public Norwegian workforce. Other results from the WIRUS project are published elsewhere [26–31].

### Sample

Employees from twenty companies were invited to participate in the study. The companies had a total of 18,000 employees, between 91 and 3,714 employees (median = 337 employees), and represented 7 branches (accommodation/food service activities, education, human health/social work activities, manufacturing, public administration/services, transportation/storage, and other service activities), as defined by the European Classification of Economic Activities [32]. Twelve companies were from the public sector, and eight were from the private sector. The recruitment procedure were developed by University of Stavanger. The companies were recruited by the Drug and Alcohol Competence Center, Rogaland in cooperation with three Occupational Health Services (OHS), accredited by the Norwegian Labour Inspection Authority. To be included in the present study, individuals were required to have status as employee (blue, white or pink collar worker, or manager, i.e., salaried person), and have a basic understanding of the Norwegian language.

Table 1 gives an overview of the participating employees. The final study sample, who had valid scores on all employed variables, consisted of 4,487 employees representing 24.9% of the eligible sample ($n$ = 18,000). The sample mean age was 45.2 years (range 18–71 years); a higher proportion were women ($n$ = 3,003, 66.9%); and a higher proportion had higher education ($n$ = 3,290, 73.3%).

**Table 1. Characteristics of the study sample and their distribution across the four job types (n = 4,487).**

| Variables | Sample | High-strain job [High demand Low control] | Low-strain job [Low demand High control] | Active job [High demand High control] | Passive job [Low demand Low control] |
|---|---|---|---|---|---|
| *All employees* [n (%)] | 4487 | 690 (15.4) | 1277 (28.5) | 915 (20.4) | 1605 (35.8) |
| *Age group* (p)[1] | | < 0.05 | 0.43 | < 0.01 | < 0.001 |
| ≤ 30 years [n (%)] | 569 | 101 (17.8) | 145 (25.5) | 128 (22.5) | 195 (34.3) |
| 31–40 years [n (%)] | 989 | 161 (16.3) | 289 (29.2) | 222 (22.4) | 317 (32.1) |
| 41–50 years [n (%)] | 1359 | 223 (16.4) | 395 (29.1) | 295 (21.7) | 446 (32.8) |
| 51–60 years [n (%)] | 1149 | 156 (13.6) | 335 (29.2) | 199 (17.3) | 459 (39.9) |
| ≥ 61 years [n (%)] | 421 | 49 (11.6) | 113 (26.8) | 71 (16.9) | 188 (44.7) |
| *Gender* (p)[1] | | < 0.01 | < 0.001 | 0.09 | 0.78 |
| Male [n (%)] | 1484 | 195 (13.1) | 473 (31.9) | 281 (18.9) | 535 (36.1) |
| Female [n (%)] | 3003 | 495 (16.5) | 804 (26.8) | 634 (21.1) | 1070 (35.6) |
| *Education level* (p)[1] | | < 0.001 | < 0.001 | < 0.001 | < 0.001 |
| Primary/lower secondary school [n (%)] | 123 | 26 (21.1) | 23 (18.7) | 10 (8.1) | 64 (52.0) |
| Upper secondary/high school [n (%)] | 1074 | 191 (17.8) | 225 (20.9) | 105 (9.8) | 553 (51.5) |
| University/college ≤ 4 years [n (%)] | 1526 | 248 (16.3) | 446 (29.2) | 314 (20.6) | 518 (33.9) |
| University/college > 4 years [n (%)] | 1764 | 225 (12.8) | 583 (33.0) | 486 (27.6) | 470 (26.6) |
| *Job size* (p)[1] | | 0.47 | 0.09 | < 0.001 | < 0.001 |
| Less than full-time job [n (%)] | 917 | 148 (16.1) | 240 (26.2) | 135 (14.7) | 394 (43.0) |
| Full-time job or more [n (%)] | 3570 | 542 (15.2) | 1037 (29.0) | 780 (21.8) | 1211 (33.9) |
| *Job position level* (p)[1] | | 0.12 | 0.46 | < 0.001 | < 0.001 |
| Regular employee [n (%)] | 3674 | 566 (15.4) | 1051 (28.6) | 609 (16.6) | 1448 (39.4) |
| Middle management [n (%)] | 732 | 118 (16.1) | 199 (27.2) | 263 (35.9) | 152 (20.8) |
| Top executive [n (%)] | 81 | 6 (7.4) | 27 (33.3) | 43 (53.1) | 5 (6.2) |
| *Working hours* (p)[1] | | < 0.001 | < 0.001 | < 0.05 | < 0.05 |
| Ordinary daytime without weekends [n (%)] | 3177 | 418 (13.2) | 970 (30.5) | 623 (19.6) | 1166 (36.7) |
| Other working arrangements [n (%)] | 1310 | 272 (20.8) | 307 (23.4) | 292 (22.3) | 439 (33.5) |
| *Workplace-boundedness* (p)[1] | | < 0.01 | < 0.001 | < 0.001 | < 0.001 |
| Works only at the workplace [n (%)] | 1237 | 156 (12.6) | 436 (35.2) | 353 (28.5) | 292 (23.6) |
| Works also outside the workplace [n (%)] | 3250 | 534 (16.4) | 841 (25.9) | 562 (17.3) | 1313 (40.4) |

[1]Significance values derived from Chi-square tests of sample proportions within job types.

## Data collection

Employees were invited to participate in a web-based study which entailed completing a questionnaire related to among other variables the employees' sociodemographic background, their jobs, and their perceptions of the work environment.

## Measures

**Sociodemographic variables.** The study included age, gender, and education level (primary/lower secondary school; upper secondary/high school; university/college ≤ 4 years; university/college > 4 years). In the logistic regression analyses, age was recoded to reflect ten-year age bands.

**Job descriptors.** Job size was registered as a continuous variable registered as percent of full-time work and recoded into a variable with three categories: less than full-time position, full time position, and more than full-time position. In the logistic regression analyses, job size was dichotomized into two categories: full-time job or more, versus less than full-time job. Job

position level was registered as regular employee, middle management, or top executive. Working hours was categorized as 'daytime work hours without work during weekends' versus all other categories. The other categories included daytime work hours with work during weekends; evening work hours with and without work during weekends; night-time work hours with and without work during weekends; shiftwork daytime/evening with and without work during weekends; shiftwork daytime/evening/night-time with and without work during weekends; and other. Finally, the employees were asked to indicate whether their job included working outside their workplace or not.

**Work demands and decision latitude.** Aspects of the work environment were measured with a short version of the Job Content Questionnaire [8, 11]. Decision latitude, a concept comprising the employee's own control over decisions in the workplace (decision authority), and the possibility of developing and using personal skills in the job (skill discretion), was measured with the sum score of the nine relevant items (Cronbach's $\alpha$ = .76). Psychological work demands, such as having an unreasonable great workload or not having enough time to get the work done, was measured with the sum score of five items (Cronbach's $\alpha$ = .75). Four items with reversed scoring were recoded prior to analysis.

## Data analysis

In accordance with the analytic strategy outlined by Karasek and colleagues [8, 11], four categories of psychosocial work environment were constructed by combining the decision latitude and psychosocial demands variables. On each variable, the sample was divided into two categories; the categories representing the median value or lower versus higher than the median value. By combining the values on each of the dichotomized variables, four job types emerge. These are commonly described as active jobs (combination of high demands and high decision latitude), high-strain jobs (combination of high demands and low decision latitude), passive jobs (combination of low demands and low decision latitude) and low-strain jobs (combination of low demands and high decision latitude).

The four defined job types were cross-tabulated with the employees' sociodemographic characteristics and job descriptors. Differences in proportions of job type between categories of sociodemographic and job descriptor variables were investigated using Chi Square tests. Logistic regression analysis was conducted to assess associations between sociodemographic and job descriptor variables and high-strain, low-strain, active and passive job (outcome variables in separate analyses). First, single logistic regressions were conducted to establish measures of the unadjusted associations, using high-strain, low-strain, active and passive job as dependent variable in separate models. For each model the reference group was the remainder of the job type (e.g. high-strain job versus low-strain/active/passive jobs). Second, to adjust for possible covariance between the independent variables, a series of multiple logistic regression models was constructed. Each of the models included as independent variables (i) individual background variables (age band, gender, education level) and (ii) work-related variables (job size, job position level, working hours, workplace-boundedness). Effect sizes were reported as odds ratio (OR) with corresponding 95% confidence intervals (CI). The association was regarded as significant when the *p*-value was lower than 0.05.

## Ethics

Ethical approval for conducting the study was granted by the Regional Committee for Medical and Healthcare Research in Norway (no. 2014/647). The employees were informed about the study's aim and confidentiality and were assured that participation was voluntary. All employees provided written informed consent to participate.

# Results

## Job types in the sample

The sample characteristics and their distribution within each of the four job types are displayed in Table 1. High-strain jobs were reported by 690 employees (15.4%), while low-strain jobs were reported by 1,277 employees (28.5%), active jobs by 915 employees (20.4%), and passive jobs by 1,605 employees (35.8%).

## Associations with high-strain jobs

As displayed in Table 1, having a high-strain job was more frequent among those in the youngest age group (17.8%), compared to those over the age of 60 years (11.6%). It was more frequent among women (16.5%) than among men (13.1%), and more frequent among those with primary/lower secondary school (21.1%), compared to those with the highest education level (12.8%). Having a high-strain job was also more frequently occurring among those whose working hours deviated from ordinary daytime work (20.8%), compared to those who worked only daytime with no work during weekends (13.2%). It was also more frequent among those whose work was not confined to the workplace (16.4%), compared to those who only worked in the regular workplace (12.6%).

The results from the unadjusted and adjusted logistic regression analyses are displayed in Table 2. In the adjusted model, four factors were significantly associated with having work classified as a high-strain job compared to the remaining job types: female gender (OR: 1.36, $p < .01$), lower education level (OR: 0.84, $p < .001$), work hours deviating from the main pattern of daytime work hours with no work during weekends (OR: 1.69, $p < .001$), and doing work also outside the regular workplace (OR: 1.24, $p < .05$).

## Associations with low-strain jobs

As displayed in Table 1, having a low-strain job did not vary significantly by age group. Having a low-strain job was more frequent among men (31.9%) than among women (26.8%), and was more frequent among those with the highest education level (33.0%), compared to those with

**Table 2. Associations between employee and job characteristics, and reporting high-strain, low-strain, active and passive jobs (n = 4,487).**

| Independent variables | High-strain job OR[1] (OR[2]) [95% CI[3]] | Low-strain job OR[1] (OR[2]) [95% CI[3]] | Active job OR[1] (OR[2]) [95% CI[3]] | Passive job OR[1] (OR[2]) [95% CI[3]] |
|---|---|---|---|---|
| Age increase in 10 years | 0.90 (0.93) [0.87–1.01] | 1.02 (1.02) [0.96–1.08] | 0.88 (0.84***) [0.78–0.90] | 1.14 (1.14***) [1.07–1.21] |
| Gender | 1.31 (1.36**) [1.13–1.65] | 0.78 (0.77***) [0.67–0.89] | 1.15 (1.36***) [1.15–1.61] | 0.98 (0.85*) [0.74–0.98] |
| Education level | 0.82 (0.84***) [0.76–0.92] | 1.33 (1.28***) [1.18–1.40] | 1.75 (1.60***) [1.45–1.78] | 0.61 (0.67***) [0.62–0.72] |
| Job size | 0.93 (1.24) [1.00–1.54] | 1.16 (0.92) [0.77–1.09] | 1.62 (1.23) [0.99–1.52] | 0.68 (0.86) [0.73–1.02] |
| Job position level | 0.94 (1.10) [0.90–1.35] | 0.99 (0.84*) [0.72–0.98] | 2.66 (2.68***) [2.28–3.14] | 0.39 (0.41***) [0.34–0.49] |
| Working hours | 1.73 (1.69***) [1.41–2.02] | 0.70 (0.73***) [0.62–0.85] | 1.18 (1.42***) [1.19–1.68] | 0.87 (0.75***) [0.65–0.87] |
| Workplace-boundedness | 1.36 (1.24*) [1.01–1.52] | 0.64 (0.72***) [0.62–0.83] | 0.52 (0.72***) [0.61–0.85] | 2.20 (1.67***) [1.42–1.95] |

[1]Unadjusted odds ratio. [2]Adjusted odds ratio. [3]95% CI interval related to the adjusted odds ratio. Adjustments were made for age (lower is reference), gender (male is reference), education level (lower is reference), job size (less than full time job is reference), job position level (lower position is reference), working hours (ordinary working hours with no work during weekends is reference) and workplace-boundness (no work outside the workplace is reference). Significance values relate to the adjusted analyses.

*$p < 0.05$

**$p < 0.01$

***$p < 0.001$.

primary/lower secondary school (18.7%). Having a low-strain job was also more frequently occurring among those who worked only daytime with no work during weekends (30.5%), compared to those whose working hours deviated from ordinary daytime work (23.4%). It was also more frequent among those who only worked in the regular workplace (35.2%), compared to those whose work was not confined to the workplace (25.9%).

Adjusting for all included variables, as shown in Table 2, five factors were significantly associated with having work classified as a low-strain job compared to the remaining job types: male gender (OR: 0.77, $p <$ .001), higher education level (OR: 1.28, $p <$ .001), lower job position (OR: 0.84, $p <$ .05), not having work hours deviating from the pattern of daytime work hours with no work during weekends (OR: 0.73, $p <$ .001), and not doing work also outside the regular workplace (OR: 0.72, $p <$ .001).

## Associations with active jobs

As displayed in Table 1, having an active job was more frequent among those in the youngest age group (22.5%), compared to those over the age of 60 years (16.9%). It was more frequent among those with the highest education level (27.6%), compared to those with primary/lower secondary school (8.1%). Having an active job was also more frequently occurring among who had at least a full-time job (21.8%) compared to their counterparts with less than full-time jobs (14.7%), and was more frequent among top executives (53.1%) and those holding middle management positions (35.9%) compared to regular employees (16.6%). Having an active job were more frequent among those whose working hours deviated from ordinary daytime work (22.3%), compared to those who worked only daytime with no work during weekends (19.6%). It was also more frequent among those who only worked in the regular workplace (28.5% %), compared to those whose work was not confined to the workplace (17.3%).

Adjusting for all included variables, as displayed in Table 2, six factors were significantly associated with having work classified as an active job compared to the remaining job types: lower age (OR: 0.84, $p <$ .001), female gender (OR: 1.36, $p <$ .001), higher education level (OR: 1.60, $p <$ .001), higher job position (OR: 2.68, $p <$ .001), having work hours deviating from the pattern of daytime work hours with no work during weekends (OR: 1.42, $p <$ .001), and not doing work also outside the regular workplace (OR: 0.72, $p <$ .001).

## Associations with passive jobs

As displayed in Table 1, having a passive job was more frequent among those in the oldest age group (44.7%), compared to those in the younger age groups (32.1% - 39.9%). It was more frequent among those whose education level was at the primary school/high school level (51.5–52.0%), compared to those with university/college education (26.6% - 33.9%). Having a passive job was also more frequently occurring among those who had less than full-time jobs (43.0%), compared to those who had at least a full-time job (33.9%), and was more frequent among regular employees (39.4% %) compared to those holding middle management positions (20.8%) and top executives (6.2%). Passive jobs were more frequent among those whose who worked only daytime with no work during weekends (36.7%), compared to those working hours deviated from ordinary daytime work (33.5%). It was also more frequent among those whose work was not confined to the workplace (40.4%), compared to those who only worked in the regular workplace (23.6%).

Adjusting for all included variables, as displayed in Table 2, six factors were significantly associated with having work classified as a passive job compared to the remaining job types: higher age (OR: 1.14, $p <$ .001), male gender (OR: 0.85, $p <$ .05), lower education level (OR: 0.67, $p <$ .001), lower job position level (OR: 0.41, $p <$ .001), no work hours deviating from the

pattern of daytime work hours with no work during weekends (OR: 0.75, $p < .001$), and doing work also outside the regular workplace (OR: 1.67, $p < .001$).

## Discussion

The aim of this study was to examine associations between sociodemographic characteristics and job descriptors and having high-strain, low-strain, active and passive jobs. For the most part, and in line with JDC theory [33], the associations indicated that the high-strain and low-strain job types are negative reflections of each other, as are also the active and passive job types.

The results suggest that employees' socioeconomic status (SES), as indicated by job position and education level, may be related to their perception of the job's demand and decision latitude. Based on the well-established associations between job type, in particular the high-strain type, and health, sick leave and well-being outcomes [12–16, 24, 25], job type may be viewed as a factor intervening between SES and desirable outcomes. Higher levels of education generally enable people to obtain a favorable position on the labor market, with better opportunities for employment characterized by meaningful tasks, high degree of autonomy and development of desired skills [34]. As indicated from our results, the possibility of framing the job with daytime only work hours, and without the burden of unwanted job travels, may also buffer against stress.

Higher age was significantly associated with higher odds of having a passive job (and lower odds of having an active job). This indicates that perceptions of demands in the workplace, but also perceptions of the skills asked for in the job and personal control over decisions regarding the work situation, are lower among older employees. While lower perceived demands may be a result of having more years of work routine by which to manage demands, the result indicates that older employees may be at higher risk of skill atrophy and demotivation in the job, which are risks associated with the passive job type [33].

Compared to men, women had higher odds of reporting a high-strain job. This finding aligns with previous research. Karasek and co-workers [11] summarized that there exist "enduring sex differences in both authority and opportunities to use and develop skills in the workplace across all the countries studied" (p. 346). Particularly within the lower SES categories, women have been found to have lower degrees of freedom at work compared to men [34]. In Norway, Sanne and co-workers [25] found that 15.2% of the women had high-strain jobs compared to 9.8% of the men, demonstrating a gender difference fairly similar to our results (16.5% vs 13.1%). In view of high demands and low control being consistently associated with adverse outcomes such as stress-related disorders and mental illness [12, 24], women may therefore be at higher risk of experiencing adverse health-related outcomes due to the higher proportion experiencing the job to be high-strain. For Norway, a country where gender equality is normatively prescribed in most areas of life [35], the results might indicate that the ideal of gender equality in work life is not fully translated into practice. This difference might also (partly) cover a reporting difference between men and women. That is, compared to men, women may be more open to express problematic aspects of their work environment.

On the other hand, it should also be noted that women had higher odds of having an active job (and lower odds of having a passive job), compared to men. Active jobs, described as contributing to employees' development, health and participation in the workplace, pose high demands while the employee has a high degree of decision latitude. Thus, women may be more inclined to perceive the working environment as posing high demands, compared to men. However, women's experiencing the job to be high-strain or active would depend on their how they perceive their decision latitude.

Each increase in educational level decreased the odds of having a high-strain job. More strongly, the increase in educational level increased the odds of having an active job (and decreased the odds of having a passive job), indicating that employees' education levels are more important for their perceived decision latitude, compared to the demands experienced in the job. Previous research in Norway has found higher levels of education to predict continued employment in a three-year perspective [36], and considering one's job to be secure may in itself be associated with lower job strain. Moreover, as exemplified in a study of nurses, having further education has been associated with higher job satisfaction and stronger intentions to stay in the job [37]. Thus, security for-, satisfaction with- and dedication to the job may all be related to higher education levels and the greater sense of autonomy and development in the job that frequently accompany increased competence.

Employees whose working hours deviated from the 'daytime only, no work during weekends' pattern had higher odds of reporting a high-strain job, while they also had higher odds for reporting an active job. Several studies corroborate our results by providing evidence that irregular work hours in terms of shift work is associated with poorer health and well-being among employees [38–40]. In fact, for employees who would prefer a regular daytime work schedule, the irregular work hours may be an example of experiencing low control over decisions relevant for the job, increasing the risk of job strain if or when demands increase. On the other hand, working irregular work hours might also reflect high personal control over one's own work schedule, indicative of an active job. Thus, the significance of working irregular work hours for reporting a high-strain versus an active job type needs to be interpreted in view of the employee's perceived decision latitude.

While being able to work from home or elsewhere outside the regular workplace might be considered a benefit, indicating increased flexibility, this interpretation is not supported from our study. Rather, working away from the ordinary work space appear to increase the risk of experiencing the job as either high-strain or passive. In both job types, decision latitude is low, whereas the perceived demands differ between the job types. While working irregular hours may be perceived as high-strain or active, depending on the level of employee decision latitude, a job requiring working away from the regular workplace is largely reflected in employees' reporting high-strain or passive jobs, both of which associated with low decision latitude.

## Study strengths and limitations

The sample size of the study was considerable and the employees were recruited from a wide variety of private and public enterprises. However, the response rate was low (24.9%) and the study sample was, compared with the entire Norwegian workforce, characterized by an over-representation of older, highly educated, and female employees. On the other hand, our sample was, with regard to distributions of gender and educational attainment, quite representative for public and state sector employees. Moreover, comparisons between the study sample and the invited sample (all employees in the included companies) revealed only minor differences regarding distributions of age and gender. Study selection analyses are presented in Appendix 1 (Supplementary file). Even though study selection bias may be conceived as a greater threat to prevalence estimates than to exploration of associations between variables [41], generalizations should be done with caution. Based on results from the study selection analyses, one may argue that our findings are, in particular, valid for public sector employees.

The cross-sectional design precludes the identification of causal associations. For example, while shiftwork may cause higher job-strain, it could also be that those who seek and eventually fill these positions are more prone to experiencing low control and high demands in relation to their work. Moreover, owing to a lack of data, the analyses were not adjusted for the

employees' occupation. Thus, for example, the association between female gender and high-strain job might be partly explained by the nature of the specific occupations commonly held by women. The adjusted associations with job types are therefore not likely fully independent associations. In future studies, the inclusion of a broader set of independent variables may contribute towards an increased understanding of the factors associated with job types.

Four subsequent statistical analyses were performed, such that one should pay attention to an increased probability of detecting significant associations by chance. In the case of the current study, using the commonly employed Bonferroni correction (dividing α by the number of comparisons) [42] would essentially indicate that associations with p-levels exceeding 0.0125 should be treated with caution. In view of the results obtained in the adjusted analysis, however, such considerations would only apply to three (14%) of the detected associations across three different independent variables (Table 2). Therefore, we consider the overall results largely unchanged even with the application of the Bonferroni correction.

## Conclusion

More than one out of three employees were classified as having a passive job, while less than one of six had a high-strain job. After adjustment for sociodemographic and job descriptor variables, female gender and lower levels of education were associated with reporting having a high-strain job. Moreover, having a work schedule deviating from ordinary daytime work and sometimes doing work outside the regular workplace were associated with higher odds of being classified with a high-strain job. The study corroborates the existing knowledge relating to the role of gender and education for employment and job satisfaction, and expands on the knowledge of factors associated with having high-strain, low-strain, active and passive jobs.

Systematic efforts towards improving employees' health and safety and the quality of their workplace environments may benefit from an awareness of factors that increase the risk of having problematic job types [43], such as experiencing a high-strain job. By being able to identify factors associated with having different types of jobs, the prevention of employees' health problems and work disability may become be more targeted. Taking the results of this study into account, prevention measures targeting the group and system levels may be more appropriate than measures targeting individual employees. In particular, companies that to a large extent employ women and persons with lower education, and–more importantly–have employees on shift work and doing work outside the workplace should place a particular emphasis on preventing adverse consequences. Preventive efforts should preferably be adapted to the needs and characteristics of the specific workplace, and in collaboration with occupational health services familiar with the specific context.

## Supporting information

**S1 Appendix. Study selection analyses (comparisons between study sample, invited sample, national workforce and public sector).** [ns]Non-significant difference; [*]Significant difference ($p < .05$); [a]n = 4487; [b]n = 18000, data obtained from included companies' personell records; [c]n = 2800000, data obtained from Statistics Norway [44]; [d]n = 849620, data obtained from Statistics Norway [44]; [e]only state sector employees, n = 159389, data obtained from Statistics Norway [45]; [f]Primary/lower secondary; [g]Upper secondary; [h]University/college; [j]Differences tested with chi-square tests.
(DOCX)

## Author Contributions

**Conceptualization:** Tore Bonsaksen, Mikkel Magnus Thørrisen, Jens Christoffer Skogen, Randi Wågø Aas.

**Data curation:** Tore Bonsaksen, Mikkel Magnus Thørrisen, Jens Christoffer Skogen, Randi Wågø Aas.

**Formal analysis:** Tore Bonsaksen.

**Funding acquisition:** Randi Wågø Aas.

**Investigation:** Randi Wågø Aas.

**Methodology:** Tore Bonsaksen, Mikkel Magnus Thørrisen, Jens Christoffer Skogen, Randi Wågø Aas.

**Project administration:** Randi Wågø Aas.

**Supervision:** Randi Wågø Aas.

**Writing – original draft:** Tore Bonsaksen.

**Writing – review & editing:** Tore Bonsaksen, Mikkel Magnus Thørrisen, Jens Christoffer Skogen, Randi Wågø Aas.

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
