## [Decision Letter · Decision Letter 0]

8 Nov 2019

PONE-D-19-27574

Who reported having a high-strain job, low strain job, active job and passive job? The WIRUS Screening study

PLOS ONE

Dear Prof. Bonsaksen,

Thank you for submitting your manuscript to PLOS ONE. After careful consideration, we feel that it has merit but does not fully meet PLOS ONE’s publication criteria as it currently stands. Therefore, we invite you to submit a revised version of the manuscript that addresses the points raised during the review process.

As you can see below, two expert reviewers assessed your paper, highlighting its potential for the journal and the field of knowledge addressed, but also pointing on some major concerns that separate the manuscript from its acceptance in its current version. You will see that the list of comments is not extensive, but these issues (especially those related to the interpretation and implications of the study findings) are crucial for the editorial decision. So, please address them with all the rigor possible and explain your rationales / detail the amendments and modifications done in the rebuttal letter.

We would appreciate receiving your revised manuscript by Dec 23 2019 11:59PM. To enhance the reproducibility of your results, we recommend that if applicable you deposit your laboratory protocols in protocols.io, where a protocol can be assigned its own identifier (DOI) such that it can be cited independently in the future. For instructions see: http://journals.plos.org/plosone/s/submission-guidelines#loc-laboratory-protocols

We look forward to receiving your revised manuscript.

Kind regards,

Sergio A. Useche, Ph.D.

Academic Editor

PLOS ONE

Journal Requirements:

1. Please include captions for your Supporting Information files at the end of your manuscript, and update any in-text citations to match accordingly. Please see our Supporting Information guidelines for more information: http://journals.plos.org/plosone/s/supporting-information.

Reviewers' comments:

Reviewer's Responses to Questions

**Comments to the Author**

1. Is the manuscript technically sound, and do the data support the conclusions?

Reviewer #1: Yes

Reviewer #2: Yes

2. Has the statistical analysis been performed appropriately and rigorously? 

Reviewer #1: No

Reviewer #2: Yes

3. Have the authors made all data underlying the findings in their manuscript fully available?

Reviewer #1: Yes

Reviewer #2: Yes

4. Is the manuscript presented in an intelligible fashion and written in standard English?

Reviewer #1: Yes

Reviewer #2: Yes

5. Review Comments to the Author

Reviewer #1: The authors reported that the relationships between factors and job descriptors and four types of JCQ categories. I think this paper was valuable.

Major

As the authors described from L71, occupations are related to the four job types (active, passive, low strain or high-strain). However, occupations did not include in the adjusted models. Therefore, ORs of dependent variables did not indicate fully independent effects. The ORs may reflected how the employers with each independent variable perceived their job after being adjusted with the included variables, and characteristics of occupations which the employers with each independent variable were prone to get. For example, the ORs of women may reflect how women perceive their job after being adjusted with the included variables, and occupations which women were prone to get (nurses aids and waitresses, etc.).

L130: blue, white or pink collar worker: Why did not they include in the models? And, if the can include in the model, please describe their definitions.

Table 2: The outcome had four categories, so choosing sits statistical method seemed to be somewhat difficult. I think this analysis may be the last resort (maybe polynomial logistic regression was normal, but only one outcome category can take the reference), and Bonferroni correction should be considered (P<0.05/4=0.0125).

Minor

Table 2

Showing unadjusted OR between adjusted OR and 95%CI of aOR was difficult to understand.

Reviewer #2: The paper is well developed and reported but the question remains so what since this is just a classification analysis that seems to be significant because sample sizes are large. The bigger question would be, how would these findings help the policy makers or whoever they are intended for. The implications section is not well developed and maybe need to further enhance them.

6. PLOS authors have the option to publish the peer review history of their article (what does this mean?). If published, this will include your full peer review and any attached files.

Reviewer #1: Yes: Yasuaki SAIJO

Reviewer #2: No

---

## [Author Response · Author response to Decision Letter 0]

6 Dec 2019

Authors: Dear Editor, thank you for considering our manuscript for publication in PLOS ONE. We hope the revised manuscript adheres to the journal standards, and look forward to your response.

Editor: You will see that the list of comments is not extensive, but these issues (especially those related to the interpretation and implications of the study findings) are crucial for the editorial decision.

Authors. We have augmented the section outlining the interpretation and implications of the study findings; see revised manuscript p. 22. 

Editor: [Please also include] 

A rebuttal letter that responds to each point raised by the academic editor and reviewer(s). This letter should be uploaded as separate file and labeled 'Response to Reviewers'.

A marked-up copy of your manuscript that highlights changes made to the original version. This file should be uploaded as separate file and labeled 'Revised Manuscript with Track Changes'.

An unmarked version of your revised paper without tracked changes. This file should be uploaded as separate file and labeled 'Manuscript'.

Authors: This letter serves as our response to the editor and reviewers. The revised manuscript is uploaded in two versions, with and without highlighted changes, labeled as appropriate.

Editor: When submitting your revision, we need you to address these additional requirements. Please ensure that your manuscript meets PLOS ONE's style requirements, including those for file naming. 

Authors: We have consulted PLOS ONE’s style requirements and amended the style of the manuscript accordingly.

Editor: Please include captions for your Supporting Information files at the end of your manuscript, and update any in-text citations to match accordingly. Please see our Supporting Information guidelines for more information.

Authors: We have consulted the Supporting Information guidelines and have amended the file accordingly. Captions for the Supporting Information file are placed at the end of the manuscript, with updated and matched in-text citations.

Reviewer #1: The authors reported that the relationships between factors and job descriptors and four types of JCQ categories. I think this paper was valuable.

Authors: No response required.

Reviewer #1: As the authors described from L71, occupations are related to the four job types (active, passive, low strain or high-strain). However, occupations did not include in the adjusted models. Therefore, ORs of dependent variables did not indicate fully independent effects. The ORs may reflected how the employers with each independent variable perceived their job after being adjusted with the included variables, and characteristics of occupations which the employers with each independent variable were prone to get. For example, the ORs of women may reflect how women perceive their job after being adjusted with the included variables, and occupations which women were prone to get (nurses aids and waitresses, etc.).

Authors: Well noted by the reviewer. However, we do not have access to data concerned with the actual job title of the participants. This large scale study used data from a wide variety of public and private enterprises, so if we did have access to job titles, the number of specific occupations would be very large and impossible to use for adjustments. Instead, we have noted this lack of information as a potential limitation of the study; see p. 21.

Reviewer #1: L130: blue, white or pink collar worker: Why did not they include in the models? And, if the can include in the model, please describe their definitions.

Authors: This expression was used to convey that all types of employees, provided they received salary and had a basic understanding of the Norwegian language, were eligible for participation in the study. We have no information available that allows for categorizing employees by these terms. However, the variable ‘job position level’ (see description in the ‘Job descriptors’ section; p. 10) may address similar distinctions, and this variable was included in the models.

Reviewer #1: Table 2: The outcome had four categories, so choosing sits statistical method seemed to be somewhat difficult. I think this analysis may be the last resort (maybe polynomial logistic regression was normal, but only one outcome category can take the reference), and Bonferroni correction should be considered (P<0.05/4=0.0125). 

Authors: Well noted by the reviewer. While we acknowledge the inherent problems related to using a series of binary models instead of one multinomial model, we would argue that choosing the latter model would answer a somewhat different research question than the one posed for the current study. In this study, we have addressed a range of sociodemographic and job-related variables and their associations with each of the four job types. Alternatively, using a multinomial model would provide ORs of associations with one job type relative to the other job types. If we were to use the Bonferroni correction procedure (essentially setting the level of statistical significance at 0.0125), the larger part of the associations detected in the multiple analysis would still remain significant (Table 2). See added discussion in the limitations section; p. 21.

Reviewer #1: Table 2. Showing unadjusted OR between adjusted OR and 95%CI of aOR was difficult to understand.

Authors: We agree, and we have re-arranged the ORs/aORs in Table 2.

Reviewer #2: The paper is well developed and reported but the question remains so what since this is just a classification analysis that seems to be significant because sample sizes are large. The bigger question would be, how would these findings help the policy makers or whoever they are intended for. The implications section is not well developed and maybe need to further enhance them.

Authors: Well noted by the reviewer. We have expanded the conclusion section of the manuscript to include the study’s implications; see p. 22. We thank both reviewers and the editor for their comments.

---

## [Decision Letter · Decision Letter 1]

18 Dec 2019

Who reported having a high-strain job, low strain job, active job and passive job? The WIRUS Screening study

PONE-D-19-27574R1

Dear Dr. Bonsaksen,

We are pleased to inform you that your manuscript has been judged scientifically suitable for publication and will be formally accepted for publication once it complies with all outstanding technical requirements.

With kind regards,

Sergio A. Useche, Ph.D.

Academic Editor

PLOS ONE

Additional Editor Comments (optional):

Reviewers' comments:

Reviewer's Responses to Questions

**Comments to the Author**

1. If the authors have adequately addressed your comments raised in a previous round of review and you feel that this manuscript is now acceptable for publication, you may indicate that here to bypass the “Comments to the Author” section, enter your conflict of interest statement in the “Confidential to Editor” section, and submit your "Accept" recommendation.

Reviewer #1: All comments have been addressed

Reviewer #2: All comments have been addressed

2. Is the manuscript technically sound, and do the data support the conclusions?

Reviewer #1: Yes

Reviewer #2: Yes

3. Has the statistical analysis been performed appropriately and rigorously? 

Reviewer #1: Yes

Reviewer #2: Yes

4. Have the authors made all data underlying the findings in their manuscript fully available?

Reviewer #1: No

Reviewer #2: Yes

5. Is the manuscript presented in an intelligible fashion and written in standard English?

Reviewer #1: Yes

Reviewer #2: Yes

6. Review Comments to the Author

Reviewer #1: The authors responded all reviewers' comments appropriately.

Reviewer #2: (No Response)

7. PLOS authors have the option to publish the peer review history of their article (what does this mean?). If published, this will include your full peer review and any attached files.

Reviewer #1: No

Reviewer #2: No

---

## [Editor Report · Acceptance letter]

20 Dec 2019

PONE-D-19-27574R1 

Who reported having a high-strain job, low strain job, active job and passive job? The WIRUS Screening study 

Dear Dr. Bonsaksen:

I am pleased to inform you that your manuscript has been deemed suitable for publication in PLOS ONE. Congratulations! Your manuscript is now with our production department. 

With kind regards,

on behalf of

Dr. Sergio A. Useche 

Academic Editor

PLOS ONE